# Geriatric Dentistry Curriculum in Six Continents

**DOI:** 10.3390/ijerph17134682

**Published:** 2020-06-29

**Authors:** Irina Xavier, Ronald L. Ettinger, Luís Proença, João Botelho, Vanessa Machado, João Rua, Ana S. Delgado, José J. Mendes

**Affiliations:** 1Clinical Research Unit (CRU)-Centro de Investigação Interdisciplinar Egas Moniz (CiiEM), Instituto Universitário Egas Moniz (IUEM), 2829-511 Caparica, Portugal; jbotelho@egasmoniz.edu.pt (J.B.); vmachado@egasmoniz.edu.pt (V.M.); jrua@egasmoniz.edu.pt (J.R.); anasintradelgado@egasmoniz.edu.pt (A.S.D.); jmendes@egasmoniz.edu.pt (J.J.M.); 2Department of Prosthodontics and Dows Institute for Dental Research, University of Iowa College of Dentistry & Dental Clinics, Iowa City, IA 52242, USA; ronald-ettinger@uiowa.edu; 3Quantitative Methods for Health Research Unit (MQIS), CiiEM, IUEM, 2829-511 Monte de Caparica, Portugal; lproenca@egasmoniz.edu.pt

**Keywords:** geriatric dentistry, curriculum development, dental care for aged

## Abstract

The purpose of this study was to examine the presence of geriatric dentistry (GD) in the curricula of worldwide dental schools, and to identify and compare their curriculum content. Eighty-three dental schools (16.4% response rate), from 24 countries, in six continents, completed a 25-item online questionnaire, to assess their GD curriculum, and were included in the study. GD was a mandatory course in 56 dental schools (67.5%), an independent subject in 14 schools (16.9%), and was taught as a series of lectures in 31 schools (37.4%). Clinically, 56 dental schools (67.5%) had some form of mandatory education in GD. The type of school, location and method of presentation were not associated with greater interest in expanding their curriculum in GD (*p* = 0.256, *p* = 0.276, and *p* = 0.919, respectively, using the Chi-square test). We found GD is a curriculum requirement in most of the surveyed dental schools and is becoming more common among dental school curricula. This survey is the first study to present data from dental schools from all continents, using a web-based survey which is a resourceful, less-expensive tool to gather data.

## 1. Introduction

Population aging—the inevitable increase in the share of older persons that results from the decline in fertility and improvement in survival that characterize the demographic transition—is occurring throughout the world [1]. In 2030, older persons are expected to outnumber children under age 10 and the number of older persons is expected to double by 2050 (nearly 2.1 billion) [1]. Many of these elderly people are at a higher risk of several oral diseases mainly because age is a risk factor [2]. Furthermore, many are demanding more complex and interdisciplinary treatment rather than the simple solutions of the past such as dental extractions of the remaining teeth and complete dentures [3,4]. The social and educational characteristics of these elderly persons are different from past cohorts, as they are better educated and have higher health expectations [4].

The term geriatric dentistry (GD) emerged in the 1970′s from discussions on how to educate dental students to treat compromised older adults [5]. Since then, it has been stated that students need to be well prepared to provide care for older adults and changes need to be made to improve GD teaching [6]. In 1978, the first survey on the content of the curricula of GD in dental schools in the United States of America (USA) was published by Swoope [7]. Throughout the years, geriatric curriculum development in dental schools has varied greatly, influenced by multiple factors such as the school characteristics and the interests of the faculty [4,8]. A recent study found that 92.8% of the responding dental schools in the USA had GD as a compulsory subject in their curriculum [9]. This study also states government funded public dental schools were marginally more interested in expanding their GD syllabus [9]. Another study reported that the non-industrialized countries where the population has a shorter life expectancy and smaller elderly population have the lowest prevalence of teaching GD [10]. The more industrialized countries, such as Canada and the USA, have integrated geriatric content in most of their dental schools [6,9,11]. 

This reported disparity in teaching GD, the aging of the world’s population, and the development of new technologies and pedagogical materials raises the need to analyze how dental schools are currently teaching GD. Therefore, this global survey aimed to examine the presence of GD in the curricula of dental schools, and to identify curriculum content, to compare these findings with previous reports. 

## 2. Materials and Methods 

### 2.1. Ethical Considerations

The study was submitted to Egas Moniz Ethical Committee and was approved under protocol No. 866).

### 2.2. Participants

A web-search and consultation with country representatives of dental associations and societies, including the WDF (World Dental Federation, Geneva, Switzerland), ADEA (American Dental Education Association, Washington, D.C., USA) and DGES (Direcção Geral de Ensino Superior, Portugal’s Ministry of Education, Lisbon, Portugal) was compiled. A world directory of dental schools compiled by the WDF was also consulted. These searches resulted in a list of 782 dental schools in 102 countries. Through institutional contacts we identified and validated the e-mails of 506 dental schools (64.7%), from 65 different countries, for contact. From 782 potential schools, only 506 had contact email. After sending the questionnaire, 399 did not produce any response, and 107 schools have answered to the survey, of which 83 completed the questionnaire properly. The name of the Dental School was also asked in the questionnaire to eliminate duplicates as several e-mails were sent to the same school in an attempt to increase response rate. These results are not shown to maintain schools’ anonymity. When two responses from the same dental school were received and all answers were the same, one would be discarded as a duplicate to avoid bias. No responses from the same dental school were received with conflicting answers. 

### 2.3. Questionnaire

The questions were based on previously applied questionnaires [3,8,12] and consisted of 25 items. Each school was invited to participate by filling out an online anonymous questionnaire. Respondents were to report about their geriatric program, whether it was an undergraduate or postgraduate program. Repeated reminders were sent to non-responding schools. A maximum of five reminder e-mails were sent. All closed-ended questions offered the option to add more details where appropriate. Out of the 506 dental schools, 107 (21.1%) accepted to participate and completed the survey. From those, 24 schools were excluded because they did not identify their country of origin/name of the dental school. Thus, a final sample of 83 dental schools, from 24 countries and six continents, were included in the study with a 16.4% response rate. Answers were received from June 2018 to February 2019. 

### 2.4. Statistics

Data analysis was performed using IBM SPSS Statistics version 25.0 for Windows (Armonk, NY, USA: IBM Corp.). Descriptive and inferential statistics methodologies were applied. The Chi-square test was used to assess the association between relevant variables. A level of significance of 5% was set for all inferential analyses.

## 3. Results

### 3.1. Dental Schools Characteristics 

A total of 83 dental schools completed the questionnaire (Table 1). Twenty schools were private (24.1%) and 63 (75.9%) were publicly funded. Of the respondents, 19 (22.9%) were associate deans, 33 (39.8%) were department chairpersons, 21 (25.3%) were program directors who were directly responsible for teaching GD, while 10 (12.0%) were faculty professors with primary responsibility for teaching GD. The majority (72.3%) of the dental schools had a 5-year program for pre-doctoral education. Countries whose contacted schools did not produce any responses were not included in Table 1.

### 3.2. Geriatric Dentistry Teaching 

All the dental schools reported that they taught GD in their pre-doctoral programs. The results are shown in Table 2. The method most frequently reported was that GD was integrated into curriculum, followed by a lecture in another course. These were the only methods reported in Africa and Oceania. In terms of who was responsible for teaching the GD courses, 22 (26.5%) schools had a specific person, 47 (56.6%) were by a multidisciplinary team and 14 (16.7%) had no one in particular responsible. When asked about the qualifications of the coordinator of the GD courses, 39 (47.0%) schools reported that the faculty member responsible had had formal training in GD.

The most common finding was that multiple departments were responsible for teaching GD (didactic 42%, clinical 35%) followed by prosthodontics departments (didactic 27%, clinical 33.8%), as shown in Table 3. Teaching GD in Comprehensive care and family dentistry department was reported in North America and Asia, while no GD was reported to be taught in oral medicine, as in an earlier study [12].

Fifty-six dental schools (67.5%) reported that GD was compulsory/mandatory in their pre-doctoral clinical and didactic courses. Also, GD was taught as a specific independent course in 28 (33.7%), schools, as an organized series of presentations in other courses in the curriculum in 28 (33.7%) schools, and as occasional lectures as a part of clinical specialties in 25 (30.1%) schools. GD was only taught as part of occasional courses in 2.4% of all schools which responded to the questionnaire.

The questionnaire also asked the schools when they taught GD in their curriculum. Forty-four schools reported that it was taught to their fifth-year (52.5%) students, while 43.8% taught it in the fourth year and 22.5% to their first-year students. Four out of the nine 6-year dental school programs taught GD during the final year of dental school.

Table 4 shows how undergraduate GD programs were funded by the surveyed dental schools. In this study, 18.0% of the schools reported no funding, 11.8% had only funding for the postgraduate programs and 20.6% had support for both undergraduate and postgraduate programs funded. No postgraduate programs were reported from North America nor from Africa, and India (*n* = 8), Brazil (*n* = 6) and Mexico (*n* = 4) were the countries with most postgraduate programs. From Oceania, only New Zealand reported a federal government postgraduate program. Regarding undergraduate programs, most countries reported funding from federal government and local/state government, while Oceania did not report undergraduate program funding.

The questionnaire also asked dental schools if they were planning to expand their geriatric programs. Fifty-one percent reported that they were currently planning on expanding the teaching of GD and these dental schools were mainly located in South America. The type of school, school location and method of presentation were not found to be associated with an increased interest in expanding the curriculum in GD (*p* = 0.256, *p* = 0.276, and *p* = 0.919, respectively).

In the undergraduate dental programs, a thesis or some sort of publication was not required in all dental schools. When asked on how many thesis, undergraduate degree work, or journal article publications on gerontology, geriatrics, or geriatric dentistry have been made in the last 10 years, it was reported that there were on average 10 publications in the last 10 years, per responding dental school (*n* = 57).

### 3.3. Curriculum Content

The subjects covered in the didactic lectures and seminar courses for both compulsory/mandatory and elective/optional GD programs are listed in Table 5. The topics were classified similarly to how they were previously reported in earlier studies [4,9], i.e., the topics were divided into those that were traditional and most likely to be taught in oral medicine and prosthodontic courses (Table 5-A) and those that were more non-traditional or gerontological (Table 5-B), and would not usually be included in a dental school curriculum. There were major differences in which topics were taught, in schools in which the GD curriculum was compulsory or elective. For instance, a topic such as “oral manifestations of systemic disease” was taught in 78 dental schools, 73.1% as a compulsory curriculum and in only 26.9% as an elective curriculum.

The most popular traditional topic was “Medical problems of elderly people”, taught in 81 of the 83 responding dental schools. It was taught by 67.9% of the schools with a compulsory curriculum and in only 32.1% of the schools with an elective curriculum. The least popular traditional topic was “Nutritional problems of the aging patient”, taught in 69 dental schools, in which 62.3% was taught in schools with a compulsory curriculum and only 37.7% in those with an elective curriculum. For gerodontological topics, the most popular one was “Barriers to dental care”, taught in 70 dental schools, of which 68.6% had a compulsory curriculum and 31.4% an elective curriculum. The least popular topics were “Speech and hearing problems” and “geriatric assessment scales”, taught in only 53 surveyed dental schools. “Other topics” were taught in 22 dental schools, of which 59.1% were taught in schools with a compulsory curriculum and 40.9% in those with an elective curriculum.

In some dental schools (*N* = 51) clinical care for geriatric patients occurred at remote sites, mostly in old people’s homes (49.0%) and nursing homes (41.2%), followed by geriatric day care centers (31.4%) and satellite or community clinics (29.4%), as shown in Table 6.

Schools with required didactic courses (*n* = 56) were more likely to have required clinical components (59.0%), no trend by country/region was found. Most schools with elective didactic courses (*n* = 27) had an elective clinical education program (70.4%).

### 3.4. Comparison with Previous Reports

The analysis of GD curriculum over time and its comparison with previous reports is listed in detail in Appendix A. Specifically, the 25-item questionnaire sent to dental schools, the inclusion of GD teaching into curriculum over time, comparison of dedicated Gerodontology teaching formats, topics covered by lectures and seminars, funding of geriatric undergraduate programs, remote sites of clinical geriatric care, and department responsibility for teaching didactic and clinical GD.

## 4. Discussion

A report of the presence of GD in the curricula of some dental schools is presented. To achieve this, we contacted 506 dental schools from 65 countries to complete an online questionnaire. Eighty-three dental schools completed the questionnaire (16.4%). Briefly, our results found that GD was being taught in the curricula of all responding schools.

This report is particularly important because it is the first to compile information from dental schools across the six continents. GD was a compulsory course in 67.5% of the responding schools, compared to the 89.0–92.8% reported [9,13,14] in the USA and the 82.0% in Europe [15]. However, Kossioni et al. in 2017 [2] reported similar findings from European dental schools (52.0%).

Additionally, 62.7% of the schools were teaching GD as a combination of didactic courses and clinical rotations. These results represent a remarkable increase compared to previous surveys which have reported teaching of GD only in 15.8–28.0% of the responding dental schools [11,16,17]. Nevertheless, the teaching format of GD varies among countries and dental schools, similar to previous reports [3,4,9,10].

In 62.7% of dental schools with GD as a didactic course also had a clinical component located at an extramural site, which was higher than 26.8% reported for Europe in 2017 [2] but lower than in the USA in 1981 [12], 2003 [11], and 2012 [17], as well as in Austria, Switzerland and Germany [16] (see Appendix A). Also, 71.0% of Chilean schools had an outreach clinical rotation site [3]. There is evidence that if dental students have clinical experiences through extramural rotations it will improve their confidence to provide care and improve the chances that they will care for frail older adult patients after graduation [18,19,20]. In this study, a clinical component was reported in 66.3% of the schools, which was higher than the 61% described in a previous European study [15]. Mandatory/compulsory clinical experiences for dental students in the USA increased from 59% in 1974 to 67% in 2003 [17] and to 67.9% in 2016 [9], compared with 71.4% reported in our study. It should be noted, however, that not all USA dental schools responded to the survey, and from the five responses from American schools, two responded their clinical education program in GD was elective.

With regard to the content of the syllabus in GD, the results of the past four decades in the USA [9], show a tendency of teaching traditional topics, associated with oral medicine and prosthodontics. These topics include, “medical problems of elderly people”; “oral and dental tissue changes associated with aging”; and “oral manifestations of systemic disease”. Other important subjects such as “barriers to dental care”, “socio-economic problems”, “demographic distribution of elderly people” and “oral management of the functionally independent adult” have also been taught over the last 40 years.

When we asked the dental schools about their intention to expand their GD curriculum, 51.0% of dental schools reported that they were currently planning to expand, which was similar to 47.1% of European schools [2] and 53.6% of USA schools [4] reported in previous studies. We found that clinical geriatric care was being delivered by students at remote sites, 49.0% in old people’s homes and 41.2% in nursing homes, similarly to previous reports [6,9,17]. As stated earlier, students who spend more time with older adults during dental school expressed more confidence in their knowledge in how to treat frail older adults and were more likely to treat them after graduation [18,19,20].

In 2017, it was found that prosthodontics, preventive and community and family dentistry departments were mainly responsible for teaching these courses in the USA [4], Austria, Switzerland and Germany [16]. Most of the respondents to our survey stated that multiple departments were responsible for teaching didactic and clinical GD, followed by prosthodontics (Table 3). In the USA, 40.0% of American dental schools reported having special patient care clinics and 18.0% required extramural rotations for their students [14]. With regard to specialization, 33.3% of the respondents to a 2017 European study reported that there were dentists specializing in Gerodontology in their country [2].

We asked dental schools if the geriatric program (whether it was an undergraduate or a postgraduate program) was funded. In this study, most undergraduate clinical programs were being funded by their federal governments, while postgraduate programs were mostly being funded by corporate/private grants (Table 4). Patient revenues accounted for only a small proportion of the funding (26.5%). Funding of postgraduate programs was less common by the government (7.4%) than by private grants (11.8%). Patient revenues made up for some of the funding (10.3%), which was not the case in earlier USA studies [12,17]. Past studies reported a high percentage of funded postgraduate programs in GD, 91.9% in Europe [2] and 60.4% in the USA [4].

Regarding research and publications about GD in the undergraduate programs, it was reported an average of 10 publications in the last 10 years, per responding dental school (*n* = 57). An earlier study reported a steady increase in the average number of annual publications in Austria, Switzerland and Germany over the years [16]. The average number of publications had risen from 3.3 in 2004, to 4.6 in 2009, to 5.5 in 2014 [16]. Research in GD may be used as a tool for facilitating learning in undergraduate programs and share valuable information, as well as increase public awareness in the subject of research.

Changes in attitudes towards GD seem to have been occurring worldwide [10]. Brazil was the first country to recognize the specialty of GD in 2001 [10]. New Zealand and Australia have developed graduate programs in special needs dentistry that includes geriatrics, as have England and Ireland [10]. The Special Care Dentistry Association in the United States have developed a diplomate program in geriatric and special needs dentistry [10].

Therefore, teachers and clinicians should be required to care for the oral health needs of the aging population, and students should be prepared by having clinical experiences in locations outside the protected environment of dental schools. Extramural programs offer students the opportunity to develop the ability to be effective in a different and challenging environment, while gaining experience in an interesting way.

### Limitations

Non-responding schools can cause significant bias and error, as we had only a fairly low response rate (16.4%) from 506 dental schools in 65 countries. We also included non-industrialized countries in our study, with less probability of teaching GD, which led to fewer responses. We decided to keep these data to raise awareness about the need to develop GD even in populations where the elderly have shorter life spans, to improve quality of life at any age.

Another limitation of this study was the lack of information of how many dental schools do not teach GD. Moreover, we could not find an accurate complete listing of all dental schools in the world, and the many schools not included in the survey may have contributed to the bias in our data. Also, the questionnaire was written in English and this could have been a barrier in countries where the English language is not widely spoken, and where school faculty members may have had to use translating tools.

## 5. Conclusions

Considering the limitations of this brief survey, it is possible to conclude that each surveyed country has a different dental education system, and geriatric dentistry education will reflect this diversity. Teaching GD has become more common among the surveyed dental schools. It also seems dental education varies by length of time and content of their curriculum. Most of the surveyed dental schools had GD as a required subject in their syllabus. Our results suggest the treatment of geriatric patients by dental students is still very limited and needs to be expanded. We believe specialized dental geriatricians can be key healthcare providers and consultants for the aging population who may have different levels of dependency. More research is needed to examine the impact of teaching GD, especially the clinical care of medically compromised frail older adults. Future research may explore whether the GD teaching in its current form is sufficient to prepare dental students in treating geriatric patients, or if a geriatric specialization is needed.

## Figures and Tables

**Table 1 ijerph-17-04682-t001:** List of the number of schools by country participating in the survey.

Continents	No. of Schools Contacted	No. of Schools Responding	Public	Private	Years of Pre-Doctoral Education	Average Class Size (No. Students)	Position of Person Responsible for Teaching Geriatric Program	No. Publications in the Last 10 Years
**North America**								
Canada	12	3	3	0	4	35–60	Multidisciplinary team.	10–40
USA	12	7	4	3	4–6	25–380	Multidisciplinary team (*n* = 3), No person in particular (*n* = 2), Director of program in geriatric dentistry (*n* = 1)	0–10
**South America**								
Brazil	66	23	19	3	5	12–140	Director of program in geriatric dentistry (*n* = 9), Multidisciplinary team (*n* = 12), No person in particular (*N* = 2)	0–100
Colombia	13	2	1	1	5	30–50	Multidisciplinary team.	2–5
Mexico	15	5	5	0	5	30–160	Multidisciplinary team (*n* = 3), Director of program in geriatric dentistry (*n* = 1), No person in particular (*n* = 1)	10–43
**Europe**								
Austria	3	1	1	0	6	40	Multidisciplinary team.	
Belgium	7	1	1	0	5	40	Director of program in geriatric dentistry.	
Croatia	2	2	2	0	6	25–100	Multidisciplinary team (*n* = 1), No person in particular (*n* = 1)	8–50
Czech Republic	5	1	1	0	5	40	Multidisciplinary team.	5
Denmark	3	1	1	0	5	70	Multidisciplinary team.	2
France	18	3	3	0	6	50–80	Multidisciplinary team (*n* = 2), No person in particular (*n* = 1)	0–3
Greece	2	1	1	0	5	130	Director of program in geriatric dentistry.	15
Italy	36	2	2	0	6	30–40	No person in particular.	0–6
Netherlands	4	1	1	0	4	75	Director of program in geriatric dentistry.	
Portugal	7	5	3	2	5	40–120	Director of program in geriatric dentistry (*n* = 3), Multidisciplinary team (*n* = 2)	0–5
Spain	10	3	0	3	5	110–242	Multidisciplinary team (*n* = 2), Director of program in geriatric dentistry (*n* = 1)	3–10
Sweden	4	1	1	0	5	80	Director of program in geriatric dentistry.	
United Kingdom	16	2	2	0	5	75–150	Multidisciplinary team.	0–1
**Africa**								
South Africa	5	4	4	0	3–5	30–60	Director of program in geriatric dentistry (*n* = 1), No person in particular (*n* = 3)	1–10
**Asia**								
India	89	10	2	8	5	40–100	Multidisciplinary team (*n* = 9), Director of program in geriatric dentistry (*n* = 1)	2–30
Israel	2	1	1	0	6	40	Multidisciplinary team.	20
Russia	10	2	2	0	5	45–300	Multidisciplinary team (*n* = 1), No person in particular (*n* = 1)	1–10
**Oceania**								
Australia	6	1	1	0	4	90	Multidisciplinary team.	22
New Zealand	1	1	1	0	5	95	Multidisciplinary team.	10

**Table 2 ijerph-17-04682-t002:** Method of didactic teaching of geriatric dentistry by percentage of total respondents (*N* = 83).

	North America	South America	Europe	Africa	Asia	Oceania
**Type of instruction**	*N* = 10	*N* = 30	*N* = 24	*N* = 4	*N* = 13	*N* = 2
Teaching geriatric dentistry	100.0	100.0	100.0	100.0	100.0	100.0
As a required course	30.0	16.7	4.3	0.0	15.4	0.0
As a specific course	0.0	6.7	21.7	0.0	30.8	0.0
Lecture in another course	20.0	3.3	21.7	25.0	0.0	50.0
Integrated into curriculum	10.0	60.0	26.1	75.0	46.2	50.0
Other	30.0	0.0	0.0	0.0	0.0	0.0

**Table 3 ijerph-17-04682-t003:** Departments responsible for teaching didactic (D) (*N* = 78) and clinical (C) (*N* = 77) geriatric dentistry.

	North America		South America		Europe		Africa		Asia		Oceania	
Department	D *N* = 10	C *N* = 10	D *N* = 29	C *N* = 28	D *N* = 21	C *N* = 21	D *N* = 4	C *N* = 4	D *N* = 12	C *N* = 12	D *N* = 2	C *N* = 2
Preventive, community, health ecology	0.0	0.0	13.8	17.9	19.0	9.5	25.0	25.0	0.0	0.0	0.0	0.0
Prosthodontics	10.0	20.0	31.0	35.7	28.9	42.9	25.0	25.0	25.0	33.3	0.0	0.0
Comprehensive care, family dentistry	30.0	40.0	0.0	0.0	0.0	0.0	0.0	0.0	0.0	8.3	0.0	0.0
Special patient care	0.0	0.0	3.4	0.0	9.5	14.3	25.0	25.0	0.0	0.0	100.0	100.0
Multiple departments	50.0	30.0	41.4	35.7	28.9	23.8	0.0	25.0	75.0	58.3	100.0	100.0
Oral medicine	0.0	0.0	0.0	0.0	0.0	0.0	0.0	0.0	0.0	0.0	0.0	0.0
Other	10.0	10.0	10.3	10.7	14.3	9.5	25.0	25.0	0.0	0.0	0.0	0.0

**Table 4 ijerph-17-04682-t004:** Funding of the geriatric undergraduate programs (*N* = 59).

Source of Funds	North America *N* = 7	South America *N* = 23	Europe *N* = 19	Africa *N* = 2	Asia *N* = 8
Federal government	14.3	52.2	84.2	50.0	25.0
Local/state government	42.9	34.8	26.3	50.0	25.0
Corporate/private grants	57.1	13.0	21.1	0.0	12.5
Patient revenue	42.9	13.0	36.8	50.0	25.0
As a special program with restricted funds	42.9	17.4	10.5	0.0	12.5

**Table 5 ijerph-17-04682-t005:** Topics covered by lectures and seminars in undergraduate programs: (A) traditional topics (oral medicine/prosthodontics) and (B) non-traditional topics (gerontological).

**(A) Traditional Topics Taught over the Past 30 Years**	**Compulsory**	**Elective**
	**(%)**	**(%)**
Medical problems of elderly people (*N* = 81)	67.9	32.1
Oral and dental tissue changes associated with aging (*N* = 80)	66.3	33.7
Drug-induced dental disease (*N* = 75)	65.3	34.7
Diagnosis and management of oral conditions of elderly people (caries, endodontics, periodontal disease, oral problems) (*N* = 80)	75.0	25.0
Oral manifestations of systemic disease (*N* = 78)	73.1	26.9
Restorative management of elderly people using modifications of standard techniques (*N* = 75)	68.0	32.0
Nutritional problems of the aging patient (*N* = 69)	62.3	37.7
**(B) Non-Traditional Topics (Gerontologic Topics Taught over the Past 30 Years** **)**		
Barriers to dental care (*N* = 70)	68.6	31.4
Socio-economic problems (*N* = 66)	65.1	34.9
Demographic distribution of elderly people (*N* = 65)	67.7	32.3
Psycho-social problems (*N* = 66)	59.1	40.9
Restorative management of persons with Alzheimer’s disease and other dementias (*N* = 60)	61.7	38.3
“Aging” and theories of aging (*N* = 68)	58.8	41.2
Oral management of the functionally independent adult (*N* = 66)	68.2	31.8
Oral management of the frail elderly (*N* = 66)	59.1	40.9
Oral management of the homebound, institutionalized and hospitalized (*N* = 63)	58.7	41.3
Visual and auditory loss (*N* = 57)	63.2	36.8
Speech and hearing problems (*N* = 53)	58.5	41.5
Restorative management of elderly people with neurological problems (Parkinson’s, myasthenia graves, etc.) (*N* = 64)	54.7	45.3
Geriatric assessment scales (*N* = 53)	50.9	49.1
Restorative management of elderly people with regards to adaptation and learning (*N* = 57)	59.6	40.4
Home care and the use of portable equipment (*N* = 55)	50.9	49.1
Other (*N* = 22)	59.1	40.9

**Table 6 ijerph-17-04682-t006:** Remote sites used for the clinical education of students in geriatric dentistry in undergraduate programs (*N* = 51).

	North America	South America	Europe	Africa	Asia	Oceania
**Sites**	*N* = 8	*N* = 18	*N* = 13	*N* = 1	*N* = 10	*N* = 1
Nursing home	75.0	44.4	46.2	0.0	0.0	100.0
Geriatric hospital/ward	50.0	27.8	30.8	0.0	0.0	100.0
Old people’s home	25.0	50.0	38.5	100.0	70.0	100.0
Geriatric day care center	25.0	55.6	23.1	0.0	0.0	100.0
Congregate meals	0.0	5.6	0.0	0.0	0.0	0.0
Clinic in a sheltered housing or high-rise apartments	12.5	5.6	0.0	0.0	0.0	0.0
Satellite or community clinics	37.5	0.0	23.1	100.0	70.0	100.0
House calls	25.0	5.6	15.4	0.0	0.0	0.0
Senior center	50.0	11.1	7.7	0.0	0.0	0.0
Recreation center	12.5	5.6	0.0	0.0	0.0	0.0
Churches	0.0	11.1	0.0	0.0	0.0	0.0
Mobile Unit	0.0	0.0	0.0	0.0	0.0	0.0

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
