# Peer review of "Geriatric Dentistry Curriculum in Six Continents"

_ijerph, 2020, doi:10.3390/ijerph17134682_

Round 1
Reviewer 1 Report
The aim of the present global survey was to establish the presence of Geriatric Dentistry (GD) teaching within dental schools, as a central topic since the worldwide increase of elderly population.
The paper is accurately presented, and the topic is current and interesting from an epidemiological point-of-view.
It could be interesting, in further research, to establish whether the GD teaching in its current form is enough to develop students able to treat the complexity (dental, local, systemic, pharmacological, etc.) of a geriatric patient, or a deep specialization is needed.
According to this reviewer’s considerations, publication of the present manuscript is strongly recommended.
Author Response
Dear Editor of the International Journal of Environmental Research and Public Health,
We are pleased with the opportunity to revise and resubmit our manuscript "Geriatric Dentistry curriculum in six continents" (Manuscript ID ijerph-837199).
Thank you for your comments, they were taken into deep consideration. Manuscript changes are highlighted in the revised manuscript. We have included the future research suggested in our manuscript as we agree is a pertinent issue to explore (page 12, line 405).
We hope that you find our responses satisfying. We hope the revised manuscript will better suit the International Journal of Environmental Research and Public Health. We are happy to consider further revisions and we thank you for your continued interest in our research.
Reviewer 2 Report
This is a well-written and researched overview of geriatric dental education, with a broad scope and many fairly obvious but interesting details to add to the literature. The following are a few minor comments:
Abstract: This sentence is confusing in an abstract and should be revised to show that this is the author’s impression based on comparison with previous work:“GD was found to be a curriculum requirement in the majority of the surveyed dental schools and seems to be becoming more common in dental school curricula.”
I would like to see some kind of a summary of the loss of dental schools: 782 to 506 not contacted, why and what schools; 506-107 not responded: this should be presented formally and summary information presented about these schools, at least at a high level; 107-83. If any extrapolation of the results could be made for these excluded schools, it would strengthen the reach of the conclusions.
I am confused about the question and reporting of publications? What does this question get at and what results and conclusions did you draw from the results?
Please further discuss and provide a brief introduction to your definitions of traditional topics versus gerontologic topics. I think I understand based on the discussion, but would like it to be better explained and defined as it is categorized as important in the results.
Author Response
Dear Editor of the International Journal of Environmental Research and Public Health,
We are pleased with the opportunity to revise and resubmit our manuscript "Geriatric Dentistry curriculum in six continents" (Manuscript ID ijerph-837199). Considering the editor and reviewers’ comments, all have been considered very important and were taken into profound consideration. Manuscript changes are highlighted in the revised manuscript. We hope that you find our responses satisfying. We hope the revised manuscript will better suit the International Journal of Environmental Research and Public Health. We are happy to consider further revisions and we thank you for your continued interest in our research.
This is a well-written and researched overview of geriatric dental education, with a broad scope and many fairly obvious but interesting details to add to the literature. The following are a few minor comments:
Abstract: This sentence is confusing in an abstract and should be revised to show that this is the author’s impression based on comparison with previous work:“GD was found to be a curriculum requirement in the majority of the surveyed dental schools and seems to be becoming more common in dental school curricula.”
Answer: We have changed the sentence as it was not clear it was the authors’ conclusion: “We found GD a curriculum requirement in the majority of the surveyed dental schools and is becoming more common among dental school curricula” (page 1, line 23).
I would like to see some kind of a summary of the loss of dental schools: 782 to 506 not contacted, why and what schools; 506-107 not responded: this should be presented formally and summary information presented about these schools, at least at a high level; 107-83. If any extrapolation of the results could be made for these excluded schools, it would strengthen the reach of the conclusions.
Answer: We detailed the flow of loss of dental schools, as suggested. “From 782 potential schools, only 506 had contact email. After sending the questionnaire, 399 didn’t produce any response, and 107 schools have answered to the survey, of which 83 completed the questionnaire properly” (page 2, line 79).
I am confused about the question and reporting of publications? What does this question get at and what results and conclusions did you draw from the results?
Answer: This question was made to appraise if Geriatric Dentistry is a speciality of interest in dental/oral research. Also, this information may be useful to establish a starting point to schools that may be interested in introducing GD as a curriculum unit and would like to start research in GD. We altered this paragraph to better convey the relevance of the number of research projects and publications in GD: “Regarding research and publications about GD in the undergraduate programs, it was reported an average of 10 publications in the last 10 years, per responding dental school (n=57). An earlier study reported a steady increase in the average number of annual publications in Austria, Switzerland and Germany over the years[16]. The average number of publications had risen from 3.3 in 2004, to 4.6 in 2009, to 5.5 in 2014[16]. Research in GD may be used as a tool for facilitating learning in undergraduate programs and share valuable information, as well as increase public awareness in the subject of research” (page 11, line 362).
Please further discuss and provide a brief introduction to your definitions of traditional topics versus gerontologic topics. I think I understand based on the discussion, but would like it to be better explained and defined as it is categorized as important in the results.
Answer: We added an explanation as to why we categorized the topics the way they were presented: “The topics were classified similarly to how they were previously reported in earlier studies[4,9], that is, the topics were divided into those that were traditional and most likely to be taught in oral medicine and prosthodontic courses (Table 5-A) and those that were more non-traditional or gerontological (Table 5-B), and would not usually be included in a dental school curriculum” (page 8, line 219).
Reviewer 3 Report
The authors presented a questionnaire-based study in order to assess the development of geriatric dentistry curricula in worldwide dental schools. In a society that is getting older, providing courses in Gerodontology is crucial to guarantee "healthy ageing".
I have few suggestions that the authors might want to take into consideration:
- Please, consider to use impersonal sentences, trying to avoid the use of personal expressions or statements.
- Please, be consistent with the use of capital letters.
- Abstract: please add the statistical method used (Line 24, pag. 1).
- Materials and Methods: please add the questionnaire as a supplementary table (Line 77, pag. 2).
- Results: please modify Table 1. I would like to suggest to list the number of dental schools by Continents and not by Countries (Line 104, pag. 3).
- Results: please correct Table 2. The sum of percentages in column "n" is not 100% (Line 118, pag. 5).
- Results: please correct Table 3. The sum of percentages in column "Didactic n (%)" is not 100% (Line 122, pag. 5-6).
- Results (Table 5): is the sum between "Compulsory" and "Elective" columns supposed to be 100%? If so, please correct Table 5 (Line 165, pag. 7).
- Results: please consider to show the results of the survey among the the different Continents and to discuss them in Discussion.
- Supplementary tables: comparing the results of previous studies with those of the present one is interesting, but it could be more (and even more correct) by comparing the results of the single Continent and not of the Continents all together.
Author Response
Dear Editor of the International Journal of Environmental Research and Public Health,
We are pleased with the opportunity to revise and resubmit our manuscript "Geriatric Dentistry curriculum in six continents" (Manuscript ID ijerph-837199). Considering the editor and reviewers’ comments, all have been considered very important and were taken into profound consideration. Manuscript changes are highlighted in the revised manuscript. We hope that you find our responses satisfying. We hope the revised manuscript will better suit the International Journal of Environmental Research and Public Health. We are happy to consider further revisions and we thank you for your continued interest in our research.
The authors presented a questionnaire-based study in order to assess the development of geriatric dentistry curricula in worldwide dental schools. In a society that is getting older, providing courses in Gerodontology is crucial to guarantee "healthy ageing".
I have few suggestions that the authors might want to take into consideration:
Please, consider to use impersonal sentences, trying to avoid the use of personal expressions or statements.
Answer: We removed personal expressions and changed some sentences throughout the manuscript to avoid personal statements (page 2, line 73; page 10, line 313).
Please, be consistent with the use of capital letters.
Answer: We revised the manuscript and corrected several acronyms, as well as several abbreviated terms (page 1, line 44; page 2, line 74; page 2, line 75; page 2, line 76; page 6, line 32; page 7, line 173; page 7, line 175; page 7, line 178; page 7, line 181; page 7, line 182; page 7, line 197; page 8, line 219; page 10, line 298; page 11, line 369; page 11, line 370; page 11, line 387; page 12, line 398; page 12, line 400; page 12, line 404).
Abstract: please add the statistical method used (Line 24, pag. 1).
Answer: We added the statistical method as suggested: “The type of school, location and method of presentation were not associated with greater interest in expanding their curriculum in GD (p=0.256, p=0.276, and p=0.919, respectively, using the Chi-square test)” (page 1, line 23). The abstract was revised in order to be kept under 200 words.
Materials and Methods: please add the questionnaire as a supplementary table (Line 77, pag. 2).
Answer: The questionnaire was added in the supplementary document and such reference has been added in the manuscript.
Results: please modify Table 1. I would like to suggest to list the number of dental schools by Continents and not by Countries (Line 104, pag. 3).
Answer: Table 1 has been modified as suggested (page 3, line 122).
Results: please correct Table 2. The sum of percentages in column "n" is not 100% (Line 118, pag. 5).
Answer: Table 2 has been completely altered to show the method of didactic teaching of geriatric dentistry by type of instruction, and by continents (page 6, line 143).
Results: please correct Table 3. The sum of percentages in column "Didactic n (%)" is not 100% (Line 122, pag. 5-6).
Answer: Table 3 was initially corrected as two decimal places were wrong, but was later completely altered to list the departments by continents (page 6, line 164).
Results (Table 5): is the sum between "Compulsory" and "Elective" columns supposed to be 100%? If so, please correct Table 5 (Line 165, pag. 7).
Answer: In Table 5, one value (33.8) was a correction of the “elective” value (33.7), placed in the wrong column. Two values were also amended because of errors in decimal places. The whole table was revised to ensure no others errors were present (page 8, line 231).
Results: please consider to show the results of the survey among the different Continents and to discuss them in Discussion.
Answer: All tables were altered to show the results by continents, with the exception of Table 5, because the size and design of the table was difficult to fit in the manuscript and it would become more complex to read the information. Several changes were made throughout the manuscript.
Supplementary tables: comparing the results of previous studies with those of the present one is interesting, but it could be more (and even more correct) by comparing the results of the single Continent and not of the Continents all together.
Answer: All supplementary tables were altered to show the results by continents, with the exception of Table S4, for the same reasons we stated for Table 5. Tables S1, S2, S3, S5 and S6 were altered to better illustrate the comparison between the results of earlier studies and our results by continents.
Round 2
Reviewer 3 Report
The authors presented a questionnaire-based study in order to assess the development of geriatric dentistry curricula in worldwide dental schools. In a society that is getting older, providing courses in Gerodontology is crucial to guarantee "healthy ageing".
Overall, the present manuscript is well-written and gives an insight into dental schools' curricula despite the fairly low response rate of Universities.
I wish to thank the authors for having accepted my suggestions and comments.